# Teratoma Assay for Testing Pluripotency and Malignancy of Stem Cells: Insufficient Reporting and Uptake of Animal-Free Methods—A Systematic Review

**DOI:** 10.3390/ijms24043879

**Published:** 2023-02-15

**Authors:** Joaquin Montilla-Rojo, Monika Bialecka, Kimberley E. Wever, Christine L. Mummery, Leendert H. J. Looijenga, Bernard A. J. Roelen, Daniela C. F. Salvatori

**Affiliations:** 1Anatomy and Physiology, Department Clinical Sciences, Faculty of Veterinary Medicine, Utrecht University, 3584 CL Utrecht, The Netherlands; 2Department of Anesthesiology, Pain and Palliative Medicine, Radboud Institute for Health Sciences, Radboud University Medical Center, 6525 GA Nijmegen, The Netherlands; 3Department of Anatomy and Embryology, Leiden University Medical Centre, 2333 ZC Leiden, The Netherlands; 4Princess Máxima Center for Pediatric Oncology, 3584 CS Utrecht, The Netherlands

**Keywords:** teratoma assay, hPSCs, pluripotency, malignancy

## Abstract

Pluripotency describes the ability of stem cells to differentiate into derivatives of the three germ layers. In reporting new human pluripotent stem cell lines, their clonal derivatives or the safety of differentiated derivatives for transplantation, assessment of pluripotency is essential. Historically, the ability to form teratomas in vivo containing different somatic cell types following injection into immunodeficient mice has been regarded as functional evidence of pluripotency. In addition, the teratomas formed can be analyzed for the presence of malignant cells. However, use of this assay has been subject to scrutiny for ethical reasons on animal use and due to the lack of standardization in how it is used, therefore questioning its accuracy. In vitro alternatives for assessing pluripotency have been developed such as ScoreCard and PluriTest. However, it is unknown whether this has resulted in reduced use of the teratoma assay. Here, we systematically reviewed how the teratoma assay was reported in publications between 1998 (when the first human embryonic stem cell line was described) and 2021. Our analysis of >400 publications showed that in contrast to expectations, reporting of the teratoma assay has not improved: methods are not yet standardized, and malignancy was examined in only a relatively small percentage of assays. In addition, its use has not decreased since the implementation of the ARRIVE guidelines on reduction of animal use (2010) or the introduction of ScoreCard (2015) and PluriTest (2011). The teratoma assay is still the preferred method to assess the presence of undifferentiated cells in a differentiated cell product for transplantation since the in vitro assays alone are not generally accepted by the regulatory authorities for safety assessment. This highlights the remaining need for an in vitro assay to test malignancy of stem cells.

## 1. Introduction

Human pluripotent stem cells (hPSCs) self-renew indefinitely and can differentiate into all cell types of the three germ layers that make up the human body. They may be derived as human embryonic stem cells (hESCs) from blastocyst-stage embryos [1] or by reprogramming somatic cells of the body to induced pluripotent stem cells (hiPSCs) [2]. Recent recommendations from the International Stem Cell Initiative (ISCI) require that all new hPSC lines (as well as any clonal derivatives) are assessed for pluripotency (by assays that depend on the ultimate applications of the cell line) [3,4].

First assessment of pluripotency often examines the expression of pluripotency-associated genes or proteins, such as *OCT3/4*, *SOX2*, *TRA-1-60*, and *SSEA3* and their comparison to reference expression profiles of validated pluripotent cell lines [5,6,7]. There are also bioinformatic tools, such as PluriTest, which are based on two decades of transcriptomic analyses on pluripotent cell lines. PluriTest compares the transcriptome of a test line with a large number of reference lines [8]. 

To confirm developmental pluripotency, differentiation towards derivatives of the three germ layers can be assessed either in vitro or in vivo. In vitro, the differentiation capacity can be demonstrated by culturing the cells as aggregates, called embryoid bodies (EBs), where they can undergo spontaneous multi lineage differentiation. Conversely, it can be assessed by inducing lineage-specific differentiation in monolayer culture through exposure to different growth factors or small molecules [9]. An alternative bioinformatic tool is ScoreCard, which can provide quantitative information of the differentiation potential of the lines by reporting the expression of germ layer-specific genes upon their directed or spontaneous differentiation [10,11].

In vivo, pluripotency can be assessed as the capacity to differentiate to mature tissues in what is referred to as a teratoma assay. This entails the injection of (pluripotent) stem cells into immunodeficient mice, where they develop tumor cell masses composed of multiple tissue types derived from the three germ layers of the embryo. These tumors are called teratomas and only form if the transplanted cells are pluripotent. However, in some cases the tumors can also contain what pathologists refer to as “embryonal carcinoma elements” and/or undifferentiated cells [12]. If these types of structure are present, the tumor has been referred to historically as a “teratocarcinoma” and the injected stem cells are considered potentially malignant [3,12].

Not all downstream applications require the evaluation of pluripotency and malignancy in vivo. To register an hPSC line in a stem cell bank such as the European Bank of Induced Stem Cells (EBiSC) or the Human Pluripotent Stem Cell Registry (hPSCreg), for example, it is sufficient to verify pluripotency based on the expression of *OCT3/4*, *SOX2*, *NANOG*, *KLF4*, *TRA1-60*, *TRA1-81*, and *SSEA4* genes or proteins [9,13]. However, the teratoma assay is still required for the assessment of the safety of hPSCs and hPSC-derived medicinal products for clinical application by the regulatory authorities. This is despite it being time-consuming, costly, and ethically questionable (due to the use of laboratory animals), as it is the only test that can be used to simultaneously evaluate both pluripotency and malignancy [3].

In view of this strong reliance on the teratoma assay, it is essential that the outcome is reliable and reproducible between laboratories and between operators and mouse strains. Proper standardization of the teratoma assay has been called for [14,15]. This systematic review therefore examined reported use of the teratoma assay in the scientific literature over the last two decades and aimed to explore (1) how teratoma assays have been conducted for the assessment of pluripotency and malignancy potential; (2) whether variables potentially influencing the reliability of the results have been standardized; and (3) if the Animal Research: Reporting of In vivo Experiments (ARRIVE) guidelines were followed [16]. The ARRIVE guidelines specify the 10 essential requirements regarding animal experiments that must be included in any manuscript to ensure the reliability of the findings, among which are details regarding the animal strain, sex, age, and number, as well as details regarding the experimental procedures such as “what”, “when”, “where”, and “why”.

## 2. Methods

### 2.1. Registration of the Study

The design and eligibility criteria were registered in the PROSPERO database (International prospective register of systematic reviews) for systematic review [17] under registration number CRD42021237843. The protocol for this systematic review was constructed according to the Systematic Review Centre for Laboratory Animal Experimentation (SYRCLE) protocol format [18] and was reviewed and approved by all authors. No deviations from the protocol occurred.

### 2.2. Search Strategy

The Patient, Intervention, Comparison, and Outcome (PICO) strategy was used as the basis to conduct a search of three scientific journal databases: PubMed, Web of Science, and Embase (Ovid), for articles where teratoma assays had been performed to assess human pluripotent stem cells, on 13 October 2020. The search covered the period from 1998, i.e., when the first hESC line was generated, until 2020 [1]. For each database, a separate search string was designed considering different operators used for combining search terms. Each string contained key words with their variations or abbreviations used in the field and thesaurus terms were used when available. The exact combinations for each database are documented in the Appendix A. The design of the search string was developed in collaboration with experts from the Waleus library at Leiden University Medical Center (LUMC).

A supplementary search was then performed in PubMed on 15 January 2021. Additionally, references of retrieved publications were screened for any relevant publications that might have been missed by the electronic search.

### 2.3. Study Selection

Study selection was performed and reported according to Preferred Reporting Items for Systematic Reviews and Meta-Analyses (PRISMA) [19], and performed in two stages with initial selection based on the title and abstract, followed by full text assessment for eligibility criteria performed by J.M.-R., M.B., and B.A.J.R.

Abstracts found after search procedures and screened publications, based on the title and abstract, were uploaded to an online database, Rayyan [20]. All records were independently screened by two reviewers, J.M.-R, M.B., B.A.J.R., or D.C.F.S., blinded for the assessment of the first reviewer. Any conflicts were discussed and, if required, resolved by a third reviewer. Publications included for full text review were exported to Endnote X7 software (Clarivate Analytics, Philadelphia, PA, USA).

Study selection was performed in two stages with initial selection based on the title and abstract, followed by full text assessment for eligibility criteria performed by J.M.-R., B.A.J.R., and M.B. The flow diagram in Figure 1 presents the procedure for literature selection.

#### 2.3.1. Inclusion Criteria

Initially, studies were included that reported in the title or abstract: (1) injection of undifferentiated human pluripotent stem cells (hPSC), (2) into any strain of immunodeficient mice, and (3) showed the results of the teratoma assay. Publications that included this information were screened in full and included for further analysis if they additionally reported the following characteristics of the assay: (4) cell type used, (5) number of cells per injection, (6) site of injection, and (7) duration of the teratoma assay.

#### 2.3.2. Exclusion Criteria

Complete publications not written in English (n = 4 written in Chinese; n = 1 written in Portuguese; n = 1 written in Bulgarian) or not presenting original findings as a full article (e.g., conference abstracts, book chapters, and protocols) and articles published before 1998 (i.e., the year of the derivation of the first reported hPSC line as hESC) were excluded from the analysis. Additional exclusion criteria were: studies that reported injection of cells differentiated from hPSCs or cells other than hPSCs, teratoma assay performed in a model organism other than immunodeficient mice, no assessment of pluripotency, studies that did not report (the results of) a teratoma assay or provided insufficient information about the assay characteristics mentioned above.

### 2.4. Data Extraction

Data regarding study characteristics and experimental set-up were extracted from the main text, figures and their legends, and Appendix A, and recorded in Excel worksheets. The data were extracted by J.M.-R., M.B. and B.A.J.R. For at least 25% of the articles, the data extracted were checked for errors by a second reviewer. Any discrepancies were discussed, if required, and resolved by a third reviewer. In accordance with our PROSPERO protocol, the following data were extracted: author, title and year of publication; cell type and passage number used, karyotyping (and method if provided), differentiation assay and additional pluripotency tests, cell number injected and pre-treatment (which might affect cell lines properties, thus assay outcome) prior to injection into the animal. In addition, cell carrier (vehicle used for cell injection) and volume injected, number of injections per mouse; mouse strain, number, sex and age of the animals used; duration of the experiment and any additional treatment of the mice, which was a part of the initial study design. The following assay outcomes were included: tumor size and tumor growth time, methodology for tumor sampling for the analysis, techniques used to assess presence of derivatives of the three germ layers. In addition, whether the histopathology analysis was performed by a pathologist, techniques and criteria used for the assessment of malignancy, and the author’s conclusion regarding the final diagnosis. In addition, experimental set-ups and methods of data analysis were compared in order to determine the impact of variability on conclusions and outcome.

### 2.5. Quality Assessment

To assess the quality of the studies included, the following indicators were examined:(1)Number of animals used per cell line;(2)Malignancy assessment; whether the histopathology reports focused on the presence of tissues derived from the three germ layers. It would be important at this point to also evaluate if the teratoma contained malignancy-related elements;(3)Additional evaluation of the haematoxylin and eosin (H&E) presented results (for example, immunohistochemistry (IHC) confirmation of the three germ layers by specific antibodies);(4)Evaluation by a pathologist; whether the histology of the teratomas required specific pathology training since these tumors are rather complex;(5)Representative histological (H&E) pictures, able to give a clear account of the tissues derived from the three germ layers;(6)Cell passage number;(7)Additional in vitro experiments for confirmation of pluripotency (for example, PluriTest/ScoreCard, additional evaluation by messenger RNA (mRNA) or transcriptome analysis, or by other means);(8)Tumor progression assessment throughout the experiment; this point is related to measurement of the tumor size, how this was approached, and planned;(9)Was animal mortality during the experiment reported, and if so, was the cause investigated and reported? If animal mortality was reported, was it related to the experiment or to other intercurrent conditions? Was the experiment shortened as a result of unexpected mortality? This information could affect the maturity of the collected tumor but also relate to monitoring of animal welfare;(10)Were there unexpected interventions related to animal welfare?(11)Was there an unexpected intervention not connected to experimental design? For example, interventions linked to administration of drugs such as painkillers or other drugs.

## 3. Results

### 3.1. Search Results

The publications included in our analysis described the use of a teratoma assay to assess pluripotency and malignancy of human stem cells over the course of just over two decades, November 1998–15 January 2021. Only references concerning hESCs and hiPSCs were included. After duplicate removal, 2193 abstracts were screened, of which 1414 (64%) were excluded based on title and abstract (Figure 1). In addition, publications that were not full-length articles or did not present original data, such as conference abstracts, reviews, or protocols, and studies that did not present any results on the teratomas were also excluded (Figure 1 and Appendix A). For the final analysis, publications were included if they stated cell type used (undifferentiated hESC, hiPSC, or both) (Figure 2A), cell number injected, site of injection, duration of the assay, and if experimental results related to the teratoma assay were reported. As a result, 492 full-text publications in total were included that met the criteria (Figure 1). Overall, in 90.0% of the publications evaluated, the teratoma assay was used to assess only pluripotency, whilst in 10.0% of the publications, the assay was used to also test malignancy (8.7% and 13.6%, respectively, for hiPSC and hESC publications) (Figure 2).

### 3.2. Differences in Experimental Set-Up of the Teratoma Assay

In the teratoma assay, a specified number of cells are injected into an immunodeficient mouse; growth of tumors is usually monitored over a period of weeks or months, and at certain time points (usually determined by tumor size), the tumor is removed and examined for the presence of germ layer derivatives and/or malignant components [21,22]. The duration of the experiment, i.e., the time teratomas were allowed to grow, varied from one week to one year. On average, tumors grown in the subcutis (under the skin) were removed after 4–32 weeks, intramuscular tumors after 1–26 weeks, and tumors grown in other organs after 1–18 weeks.

In the included studies, a relatively large variation in the numbers of injected cells, ranging from 800 to over 10 million cells per injection was observed (Figure 3A). Most studies (78%) reported the injection of around one million cells per animal (Figure 3A). Despite the number of transplanted cells being crucial to determining pluripotency, our analysis showed that the exact number was not always reported. Instead, the information was, in some cases (8.7%, n = 43), limited to the number of cells from a surface area of the culture dish with a certain (percentage of) confluency; we recalculated this to cell numbers, estimating 132.000 cells per confluent square centimetre. In other studies evaluated, only a range of cell numbers was provided without exact specification (Figure 3A). When cell numbers were not provided or could not be recalculated, for instance ,when only numbers of cell colonies were mentioned, the publication was excluded from further analysis (n = 2, 0.4%). The vehicle (medium and/or extracellular matrix) in which cells were injected, which can affect their engraftment [23] was specifically mentioned in 56.3% (n = 277) of publications. The composition of the vehicle varied depending on the publication, from a neutral buffer such as Hanks’ Balanced Salt Solution (HBSS) or Phosphate-buffered Saline (PBS), extracellular matrix (ECM) such as Matrigel^®^, or collagen injected alone or mixed with buffer or media in varying proportions (Appendix A).

The site of injection in the mouse body can affect the efficiency of tumor development and even its composition, due to the niche-specific biochemical and cellular cues [23,24]. Despite this, different mouse injection sites have traditionally been used in teratoma assays. The most common reported sites were subcutaneous (41.7%), intramuscular (23.4%), or into an organ (28.7%) (Figure 3B). In most of the reports of subcutaneous injections, the exact body location was also not defined, whilst others reported locations with variable accuracy such as “between the scapulae” or simply “the back” (Figure 3C). A similar lack of detail regarding the injection site was observed in the group where the injection was intramuscular. The majority of these publications reported the injection site simply as “intramuscular” or “rear leg muscle”, while only 5.4% of the publications with intramuscular injection reported the specific location such as “Tibialis anterior” (Figure 3D). Testis (66.7%) and kidney (34.8%) were the most frequently reported organs used as injection sites (19.1% and 10.0% of the total number of analyzed studies, respectively), whilst only a few publications used liver or heart for cell injection (Figure 3E). In most cases when the cells were transplanted into testis or kidney, it was not clear whether cells were injected directly into the organ or under the capsule of the organ. In the category “other”, we placed anatomical locations which did not fit in any of the categories described above. “Hindleg” was often (5.6%) reported as injection site without specifying whether the cells were transplanted subcutaneously or intramuscularly, and were thus included in the “other” category (Figure 3F). In addition, in 5.5% of the publications analyzed, cells were injected into multiple sites in the same mouse.

In the majority of the publications evaluated (63.2%, n = 311), the sex of the injected mice was not mentioned. When this information was provided, male mice were preferentially used (82.9%, n = 150) compared to females (17.1%, n = 31). For both sexes, age of mice used for injection was similarly poorly described, since in most reports this information was missing (59.6%, n = 293). When this information was provided, young adults (aged 4–8 weeks; 34.3%, n = 169) were most frequently used.

Over the last few decades, many strains of immunodeficient mice have been developed, allowing researchers to tailor the mouse model used to their experiments. However, this has also generated confusion when reporting the mouse strain used to make teratomas. Our analyses found that relatively few publications presented full strain details, with the majority using collective terminology such as SCID, NOD/SCID, CB17/SCID, NSG, or Nude (Appendix A). Since complete strain nomenclature was rarely used, we were not able to assess differences in results between specific mouse strains.

### 3.3. Tumor Evaluation

As expected, histology and the specific use of H&E staining of paraffin-embedded sections was the most widely used method (96.7%, n = 477) to evaluate the composition of tumors. Apart from H&E staining, IHC of tumor sections were frequently used to identify cells from different germ layers or the presence of undifferentiated cells. In publications where histological analysis was performed using H&E stainings, representative images of the sections were shown to demonstrate the presence of tissues originating from the three germ layers, illustrating pluripotency of the cells tested. In the majority (65%) of reports, this was the only analysis documented (Figure 4A). On occasion, mRNA expression measured by (quantitative) reverse transcription-polymerase chain reaction (qRT-PCR) was used for whole transcriptome analysis on parts of the tumor (Figure 4A). 

Since pluripotency can also be demonstrated with in vitro methods, we examined whether other differentiation experiments were performed besides teratoma assays. Indeed, 60.6% of the analyzed publications also reported results of monolayer or 3D differentiation in EBs, either spontaneous or directed. In all of these publications, the ability to generate derivatives of the three germ layers was clearly demonstrated using a variety of markers by IHC or mRNA levels.

In 10% of the publications analyzed, malignancy was evaluated based on H&E staining and cell morphology and/or by examining expression of marker proteins by IHC/IF (immunofluorescence) (Figure 4B). Additional tumor components were sometimes identified and classified as “immature teratoma”, “teratocarcinoma” “dysgerminoma”, or simply, “malignant tumor”. Malignant cells were detected in 32.6% (n = 16) of the publications included in our analysis that evaluated malignancy.

### 3.4. Use of the Teratoma Assay from 2000 to 2020

The problem of large variations in the experimental setup and reporting of the teratoma assay was first raised and discussed many years ago, culminating in several calls for standardization [14,15,25]. In addition, a number of alternative in vitro and in silico methods have been developed that could reduce the numbers of experiments performed. However, the number of publications that include teratoma assays has not decreased in more than 20 years (Figure 5A), although there have been some moderate changes in the experimental set-up. From 2005 onwards, cells have largely been injected subcutaneously and this has become the most frequently used injection site (40.8%, Figure 5B). The most commonly used solutions for injection have become ECM (22.4%) or ECM mix (with buffer or cultured medium, 41.3%) (Appendix A), but there is still variation in the cell suspension solution.

### 3.5. Quality Assessment

In many of the publications analyzed, the teratoma assay was presented as a ‘check box’ to demonstrate pluripotency, rather than a critically analyzed experiment. As a result, important details essential for objective analysis and transparent reporting were lacking. Only 37.6% (n = 185) of the publications analyzed mentioned the number of animals used to perform the teratoma assay. Even then, we found that a range rather than an exact number of animals had been provided; therefore, it was not always clear how many animals were used per cell line.

Other variability indicators included professional pathology assessment of the tumor. The tumors were assessed by a pathologist or “pathology unit” in only 4.7% and 2.2% of the studies, respectively. In addition, information regarding the methodology for tumor sampling was lacking.

Similarly, risk of variability derived from animal experiments was evaluated based on provided information regarding assessment of tumor progression, intervention due to welfare issues, or interventions that were not related to the experimental design. This information was consistently absent in the publications, as in all cases these were “Not reported”, which could lead to variability since these factors can highly influence the reliability and reproducibility of the procedures and the quality of the outcome (Figure 6).

## 4. Discussion

The use of mice as an experimental model for the induction of teratomas has been undertaken for at least seven decades [26,27,28,29,30,31]. Whilst the teratoma assay has been subject to scientific scrutiny and ethical questions over many years [14,26,32], it continues to be routinely used by researchers for testing pluripotency. This is despite numerous calls for its standardization, for example, by Müller et al. [14]. Nonetheless, in this study we have found that the lack of standardized reporting of key quality indicators, as well as lack of standardization of the procedure itself, continues. During this systematic review, we found that a total of 172 publications could not be included based on our criteria due to the absence of crucial details that could influence the reliability of the reported outcomes, such as number of injected cells, duration of the experiment and place of injection (Appendix A). In our study, for the initial screening, we included as a search term: “teratoma assay”, meaning that the title and abstract should include the term “teratoma assay” (See Appendix A). This served as the basis of our search strategy aiming to collect all relevant articles on the topic. We can however not exclude that there is a possibility of other articles not mentioning “teratoma assay” in the title or abstract that did perform the assay and thus were left out of our screening procedure. We believe that despite this potential limitation, the sample of articles included is representative of the published papers where the teratoma assay was used.

Our findings appear to agree with observations made by Percie du Sert et al. [16] on the lack of consistency in adherence to the ARRIVE guidelines, which set out the requirements for performing and reporting animal research. Transparent and accurate reporting is crucial to improve the reproducibility of scientific research. Such reporting facilitates researchers being able to consider methodological rigor of the studies, assess how reliable the reported findings are, and be able to reproduce or build upon such work [33]. In turn, such reporting would increase the scientific validity of the results and maximize the knowledge gained from each experimental study. The omission of essential information can lead to scientific and ethical concerns being raised, including those regarding animal welfare [34]. However, in our analysis, we found a relatively large variation in the numbers of injected cells and the graft sites chosen, which in many cases used non-specific, generic descriptions of the exact anatomical site. It is essential that such information is provided, given that any variance in such factors may influence the development of the resultant teratoma, compromising objective comparisons between different cell lines [25,35].

Such variance in the reporting and lack of adherence to the ARRIVE guidelines not only affects the ability of researchers to reproduce such methodologies but is likely also to lead to detrimental consequences for animal welfare, such as unnecessary pain and suffering in the mice used. As an example, the choice of the graft site: if the graft site is an internal organ such as kidney or testicles, it is more challenging to track and assess the size and growth of the resultant teratoma, leading to animal distress. Similarly, whilst there is a historical basis for teratomas derived from the testis, due to the discovery of spontaneous testicular teratomas in the 129 mouse strain [29], the transplantation of cells into mouse organs is more harmful to the animal than subcutaneous injections, and there is no indication that injection into an organ is more efficient than subcutaneous injection. It is therefore surprising that in recent years, cells were reported as having been transplanted into the mouse testis or under kidney capsule without reporting the humane endpoints or how tumor growth was assessed.

We also found that there were variations in the reported cell volumes injected, ranging from volumes of 600 to 1000 μL in subcutaneous injections, for example, despite evidence being available that injection of smaller volumes is equally effective [36,37,38]. Furthermore, there are published guidelines for injection volumes, where it is claimed that exceeding ten times the advised volumes causes pain and discomfort in the mouse and it should not be allowed [39]. Other commonly unreported variables that should always be disclosed are the cell passage number, as prolonged in vitro culture may lead to genomic or epigenetic abnormalities [40,41,42]; and the use of Matrigel^®^ for cell suspension, as it has been found to increase the efficiency of subcutaneous teratoma formation when co-injected with hESCs [23].

Laboratory animals used in research and education are protected by legislation. For example, in the European Union, they are protected under EU Directive 2010/63/EU following the principles of replacement, reduction, and refinement (3Rs), and highlighting that the use of animals for research purposes must only be considered when there is no non-animal method available. As such, this and similar legislation in other countries places researchers under legal obligations to use an in vitro model, if such models exist. In this regard, hPSC PluriTest and ScoreCard are validated methods that have been widely tested by the International Stem Cell Initiative (ISCI) as appropriate alternative methods for pluripotency evaluation according to the downstream application of the cells [3]. Analysis of gene expression by PluriTest can be used to rapidly screen and identify cells that also that meet the criteria of pluripotency as a status [3]; if direct and quantitative confirmation of differentiation capacity is required, the ISCI recommends in vitro spontaneous and directed EB differentiation combined with bioinformatic ScoreCard analysis [3]. 

The use of PluriTest and ScoreCard methods instead of the in vivo teratoma assay for testing pluripotency would already have a significant impact on the reduction of animal use in hPSC research, considering that pluripotency evaluation is the primary goal for the application of the teratoma assay (Figure 2). It is remarkable how, since the first publications of use of ScoreCard and PluriTest (2011 and 2015, respectively), only 50–60 articles have been published reporting their use for pluripotency assessment (numbers extracted from the same search engines used in our study). This represents an estimated mere 10–20% compared to the articles that were published since then reporting use of the teratoma assay, further highlighting the poor adherence to these alternative animal-free approaches. 

To date, the teratoma assay is the only assay that provides valuable information on the malignant potential of cells; this is relevant to the pre-clinical safety assessment of hPSCs [3,43]. Further research efforts are needed to identify in vitro genetic and epigenetic biomarkers indicative of malignant potential. In addition, the meaning of histological features such as yolk sac and immature neural elements present in the teratoma assay reported as possible malignancy features need further investigations [3].

## 5. Conclusions

Despite the legislation, the 3R principles, and enforcement of the ARRIVE guidelines, our systematic review has demonstrated that this has had little to no impact in better reporting and reduction in the number of teratoma assays being performed.

Although the teratoma assay is still considered by many as the gold standard for pluripotency and malignancy of hPSCs, the large variability in performing the assay demonstrates that it is far from standardized. Why, then, is there continued use of the teratoma assay despite its limitations and animal-free alternatives being available? It could simply be that it is still perceived by some editors and researchers as the “gold standard” [44].

Our results also indicate that, although the teratoma assay is the only test able to assess pluripotency and malignancy potential simultaneously, malignancy assessment is rarely the primary goal while animal-free methods are available for testing pluripotency of stem cells.

We hope that this systematic review will raise awareness amongst researchers and publishers of the clear need for transparent reporting and the use of veterinary guidelines for the care of laboratory animals to limit the detrimental consequences on their welfare and to improve experimental reproducibility. In addition, awareness is needed that the teratoma assay should only be used where in vitro alternatives are not available, such as for malignancy assessment, and not for pluripotency evaluation, the main pretext for this assay. Perhaps the most important lesson from this systematic review is the unmet need for in vitro assay/s which can assess malignancy potential and therefore replace entirely the need to carry out in vivo teratoma assays.

## Figures and Tables

**Figure 1 ijms-24-03879-f001:**
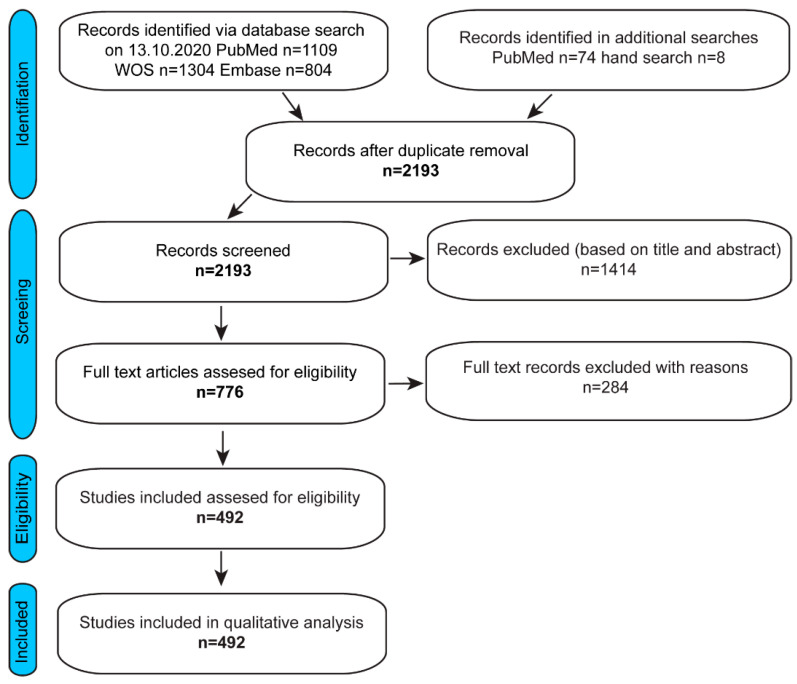
PRISMA chart. Flow diagram presenting the procedure for literature selection.

**Figure 2 ijms-24-03879-f002:**
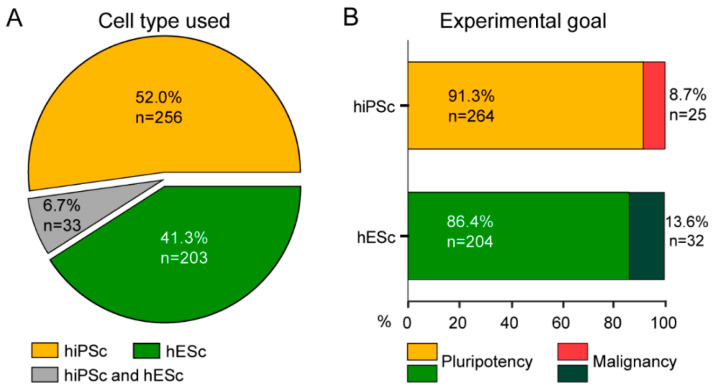
Cell types used and the primary goal of the teratoma assay. (**A**) Number of hiPSC and/or hESC cell lines used in the teratoma assay between (2000–2020). (**B**) The teratoma assay is mostly used for assessing pluripotency, and in a limited number of cases, for assessing malignancy.

**Figure 3 ijms-24-03879-f003:**
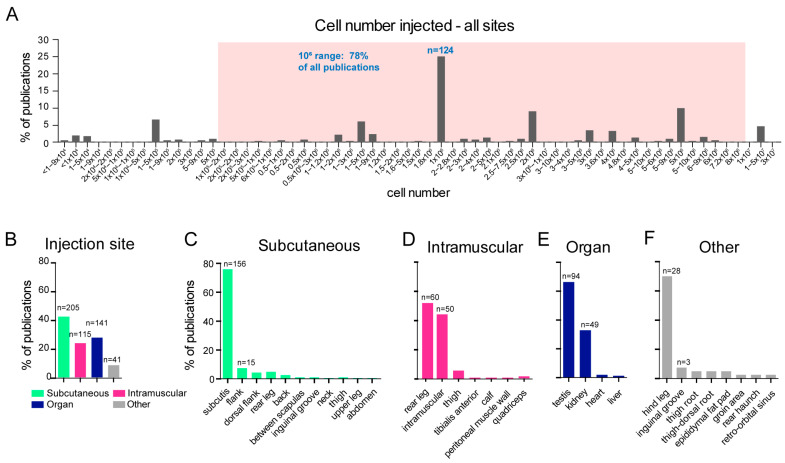
Overview of number of injected cells and anatomical injection site. (**A**) Distribution of injected cell numbers. In case surface area and confluency were reported rather than cell number, cell number was recalculated; 1 × 10^6^ represent the most frequent number of injected cells (25.2%, n = 124). The red box includes all cell numbers within the 1 × 10^6^ range. (**B**) Most commonly used anatomic areas for cell injection (subcutaneous, intramuscular, organs, other). Note that percentages do not sum up to 100% due to various reports where the assay was performed in multiple injection sites. (**C**–**F**) Reported and described anatomical sites for injection.

**Figure 4 ijms-24-03879-f004:**
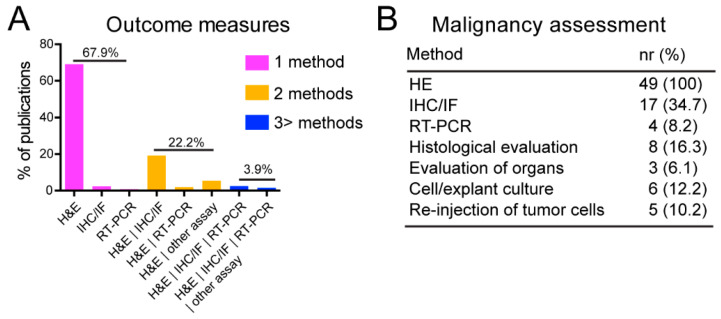
Outcome measures and malignancy assessment. (**A**) Techniques and their combinations used for tumor assessment. (**B**) Method(s) and the numbers (%) of publications assessing potential malignant components of the tumors.

**Figure 5 ijms-24-03879-f005:**
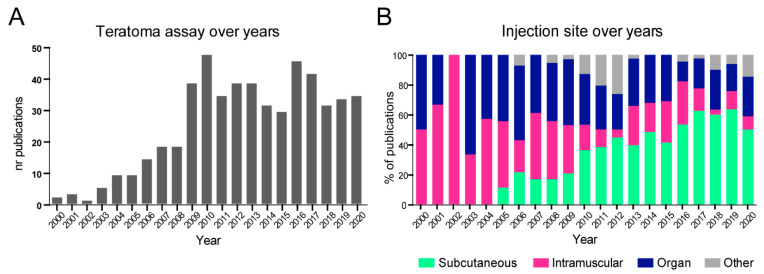
Teratoma assay over the years. (**A**) Number of publications reporting use of teratoma assay per year, from 2000 to 2020. (**B**) Overview of anatomical injection sites used to inject cells (from 2000 to 2020).

**Figure 6 ijms-24-03879-f006:**
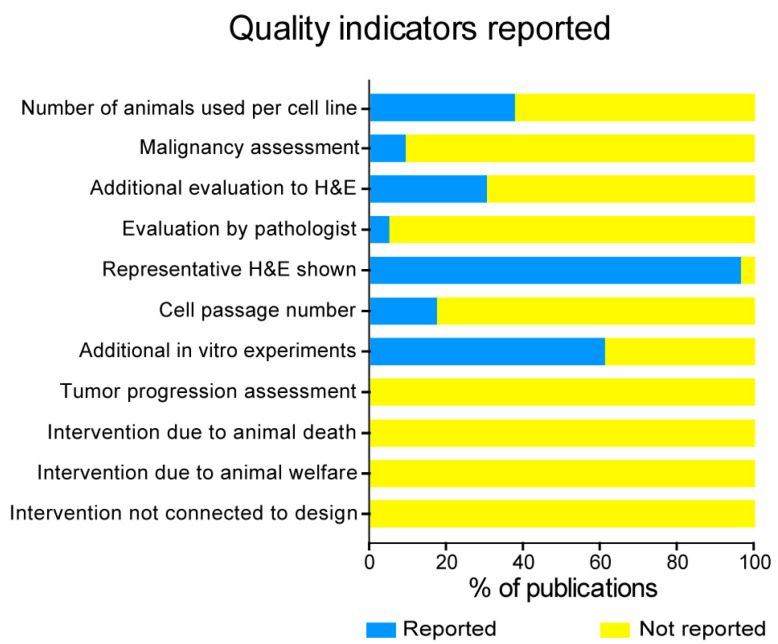
Reporting of key quality indicators. Reporting of quality indicators including animal experiment-derived variables affecting quality and variability.

## Data Availability

All data is available at request.

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
