# Peer review of "Teratoma Assay for Testing Pluripotency and Malignancy of Stem Cells: Insufficient Reporting and Uptake of Animal-Free Methods—A Systematic Review"

_ijms, 2023, doi:10.3390/ijms24043879_

Round 1
Reviewer 1 Report
The review is dedicated to describe the necessity of pluripotency and malignancy tests of iPS in vivo and in vitro based on the already published articles. The review is easy to read, big published articles analysis was done, however, it is a sociological type of review, rather than scientific, and would be better to submit it to the more generous, social-medicinal topics-related journals. The deeper description and analysis of pluripotency results would strengthen this review.
Minor remarks:
1. Figures should be in the section, where the text talks about it.
2. The line 81. The sentence is not finished.
3. Line 86 probably instead of “these specify…” should be “they specify…”
4. Line 127. The Figure 1 is usually written with the capital letter.
Author Response
REFEREE 1
The review is dedicated to describe the necessity of pluripotency and malignancy tests of iPS in vivo and in vitro based on the already published articles. The review is easy to read, big published articles analysis was done, however, it is a sociological type of review, rather than scientific, and would be better to submit it to the more generous, social-medicinal topics-related journals. The deeper description and analysis of pluripotency results would strengthen this review.
Thank you for reading our review with positive interest. In our view, this work has as target group, scientists in the field of stem cell biology; this is the reason why we have chosen this journal and this special issue. Indeed, animal experimentation has long been at the center of public and scientific debates. The 3Rs principles of good experimental practice (replacement, reduction, refinement), first formulated in 1959 have been embedded in the EU Directive 2010/63/EU with focus on replacement and guide ethical committees in their assessments of animal experiments. The teratoma assay is an animal experiment.
The same directive acknowledges that animals have an intrinsic value and clearly states that no animal experiment should be performed for scientific or educational purposes if these can be reached with animal-free methods (EU Directive 2010/63/EU/art. 12). The directive also states the animal experiments should be carried out following the highest scientific and animal welfare standard. This is why we believe this review has a deep scientific interest, rather than sociological, as it describes the current limitations of one of the most in vivo widely-used assays in iPSC/hESC research. Many articles have positioned the transition to animal-free innovations a “techno-moral” innovation and studied this topic showing that in the biomedical fields “scientific inertia” makes implementation of innovations difficult (https://doi.org/10.1016/j.shpsa.2021.06.016). As such, the impact of our study has a scientific perspective, which is to highlight the relevance of standardizing the assay as well as to aim to develop and embrace in vitro alternatives, all of which can highly impact the work done by thousands of biomedical researchers worldwide.
Minor remarks:
- Figures should be in the section, where the text talks about it.
We have now adjusted the location the of the figures closer to the sections where we refer to them. Please find them in the newest version of the manuscript.
- The line 81. The sentence is not finished.
The sentence starting in line 81 is long and therefore not clear. To make it easier to read we changed this part to: “This systematic review therefore examined reported use of the teratoma assay in the scientific literature over the last two decades and aimed to explore 1) how teratoma assays have been conducted for the assessment of pluripotency and malignancy potential; 2) whether variables potentially influencing the reliability of the results have been standardized; and 3) if the Animal Research: Reporting of In Vivo Experiments (ARRIVE) guidelines [16] were followed. The ARRIVE guidelines specify the 10 essential requirements regarding animal experiments that must be included on any manuscript to ensure the reliability of the findings, among which are details regarding the animal strain, sex, age, number, as well as details regarding the experimental procedures such as “what”, “when”, “where” and “why”.”
- Line 86 probably instead of “these specify…” should be “they specify…”
This has been changed, see point 2
- Line 127. The Figure 1 is usually written with the capital letter.
This has been changed to Figure
We have also performed the english proefreading.
Reviewer 2 Report
- For a systematic review, information of “492 included publications” need to be given.
- For the inclusion criteria, “Initially studies were included that reported in the title or abstract: (1) injection of undifferentiated human pluripotent stem cells (hPSC), ….” In some studies, teratoma assays were conducted; however, they are not introduced in the title or abstract? How do the authors deal with these studies? Are they excluded?
- In the part of “3.4 Use of the teratoma assay from 2000 to 2020”, it seems that the second paragraph is not closely related to the sub-title. In addition, is the significant increasement of publications number from 2000 to 2020 due to the development of iPSC? Maybe the author can calculate the percentage change of ESC and iPSC from 2000 to 2020?
- The teratoma assay is usually used to evaluate the pluripotency of hPSC. As the authors mentioned, other in vitro methods such as detection of pluripotent genes or proteins, hPSC ScoreCard and PluriTest are also helpful. Some studies used both teratoma assay and in vitro methods. Did the authors analysis the relationship between them? I think these information are also important.
Author Response
REFEREE 2
- For a systematic review, information of “492 included publications” need to be given.
The total number of included publications in our study is stated in line 210: “As a result, 492 full text publications in total were included that met the criteria”. Similarly, this number is shown in Figure 1, also referenced in the text after the aforementioned sentence. The list of included articles in our study can be now found in Appendix 2.
- For the inclusion criteria, “Initially studies were included that reported in the title or abstract: (1) injection of undifferentiated human pluripotent stem cells (hPSC), ….” In some studies, teratoma assays were conducted; however, they are not introduced in the title or abstract? How do the authors deal with these studies? Are they excluded?
As part of the string used for the initial screening of papers that would be relevant for our study, we included as a search term: “TIAB: teratoma assay”, meaning that the title and abstract should include the term “teratoma assay” (See Table S1). This served as the basis of our search strategy aiming to collect all relevant articles on the topic. We can however not exclude the fact that there is a possibility of other articles not mentioning “Teratoma assay” in the title or abstract that did perform the assay and thus were left out of our screening procedure. We believe that despite this potential limitation, the sample of articles included is representative of the published papers where the teratoma assay is used.
- In the part of “3.4 Use of the teratoma assay from 2000 to 2020”, it seems that the second paragraph is not closely related to the sub-title. In addition, is the significant increasement of publications number from 2000 to 2020 due to the development of iPSC? Maybe the author can calculate the percentage change of ESC and iPSC from 2000 to 2020?
Based on the reviewer’s suggestion, the second paragraph of Section 3.4 has been moved to the end of Section 3.3.
Although an increase in publication number of iPSC related paper is possible, our systematic review focused on selecting stem cells papers where the teratoma assay was used for testing pluripotency and malignancy. Whether an increase in publications is due to the development of iPSCs is hard to determine, since there is a general increase in published papers throughout the years. In addition, we believe this is not critical for our review since the subject is use of the teratoma assay, independent of the use of ES or iPS cells.
In the legend of figure 5, the part ‘Changes in teratoma experiments over the years’ has been removed
- The teratoma assay is usually used to evaluate the pluripotency of hPSC. As the authors mentioned, other in vitro methods such as detection of pluripotent genes or proteins, hPSC ScoreCard and PluriTest are also helpful. Some studies used both teratoma assay and in vitro methods. Did the authors analysis the relationship between them? I think these information are also important.
We agree on the relevance of PluriTest/ScoreCard as alternative pluripotency assessment procedures. Nonetheless, for this review we wanted to focus on the little standardization the teratoma assay, the most commonly pluripotency assessment evaluation method, presents. The relationship between the outcomes in these assays was thus beyond the scope of this systematic analysis. Other publications have convincingly demonstrated that both ScoreCard and PluriTest can be a reliable alternative for the teratoma assay to demonstrate pluripotency of cells. We refer to this in the new version of the manuscript in line 421 (No Markup setting) that there are publications from the International Stem Cell Initiative where these assays have been compared and have concluded that these are relevant and methods for testing pluripotency.
Reviewer 3 Report
In their review, authors present a comprehensive survey of teratoma assay usage and its evolution over the last 2 decades. Teratoma assay is still considered the “gold standard” regarding characterization of stem cells and their pluripotency/differentiation potential. Surprisingly, the method is still quite poorly standardized, according of the results of presented paper, albeit has been in use for several decades. Such unexpected observation deserves definitely more attention in the future and should be addressed correspondingly in future (experimental) papers (all this is nicely discussed in this paper as well).
Authors exhaustively surveyed the literature and presented the results in a clear and transparent way. Methods are described in a sufficient details allowing back-tracking of presented results.
I have some minor comments:
1) In the Fig. 3A some values are highlighted by a red box. However, I see no purpose of this box. Statements “within 10^6 range” (in figure legend) and “… around 1 million cells…” (line 225) are a little bit vague and the precise meaning/importance is not clear to me.
2) Authors mentioned the existence of alternative methods ScoreCard and PluriTest for in vitro assessment of pluripotency. It is not explicitly mentioned how frequently have been these methods used since their original description. Do authors have data on this topic. If the usage of these “alternative” methods is increasing or not? In my opinion, it would be interesting to present such data and discuss their importance (e.g. in the paragraph at lines 326-330 or in Discussion) or why these methods are (not) used as frequently as they should be.
3) English should be corrected throughout the article – I would recommend a careful checking as some sentences are not clear enough or hard to follow (focus particularly on the Abstract section). Here are some examples/typos/suggestions to correct:
-lines 15-16: the sentence is missing a verb?
-line 44: “are” instead of “be”
-line 45: should be reformulated (by assays determined by the ultimate… ?)
-line 63: “different” seems to be over-abundant
-line 68: “stem cells” or “cell lines”
-line 73: genes or proteins
-line 80: “is” instead of “be”
-line 87: 10 essential requirements
-line 196: “just” should be deleted
-line 198: duplicates
-line 270: common (or better “frequent”?)
-line 274: … for injection. (“site” should be deleted)
-line 277: “was” instead of “were”
-line 317-318: … a number of alternative … reduce the number of …
-line 335: “Changes in teratoma experiments over the years” is not clear why present here
-line 349: delete “there”
-line 364: … for at least seven …
-line 436: among
Author Response
REFEREE 3
In their review, authors present a comprehensive survey of teratoma assay usage and its evolution over the last 2 decades. Teratoma assay is still considered the “gold standard” regarding characterization of stem cells and their pluripotency/differentiation potential. Surprisingly, the method is still quite poorly standardized, according of the results of presented paper, albeit has been in use for several decades. Such unexpected observation deserves definitely more attention in the future and should be addressed correspondingly in future (experimental) papers (all this is nicely discussed in this paper as well).
Authors exhaustively surveyed the literature and presented the results in a clear and transparent way. Methods are described in a sufficient details allowing back-tracking of presented results.
I have some minor comments:
- In the Fig. 3A some values are highlighted by a red box. However, I see no purpose of this box. Statements “within 10^6 range” (in figure legend) and “… around 1 million cells…” (line 225) are a little bit vague and the precise meaning/importance is not clear to me.
The reason for adding the red box is to emphasize that in most (78% of the papers) of the studies the number of injected cells was in the range of 106 cells (106<n<107). We want to indicate that even within this range, many different cell numbers are used, further highlighting our statement in line 235-236 (No Markup setting) “Despite the number of transplanted cells being crucial to determining pluripotency, our analysis showed that the exact number was not always reported”. For the reviewer the meaning of around 1 million and in the 106 range are not clear and that is exactly why we added the red box, to make this more visual.
2) Authors mentioned the existence of alternative methods ScoreCard and PluriTest for in vitro assessment of pluripotency. It is not explicitly mentioned how frequently have been these methods used since their original description. Do authors have data on this topic. If the usage of these “alternative” methods is increasing or not? In my opinion, it would be interesting to present such data and discuss their importance (e.g. in the paragraph at lines 326-330 or in Discussion) or why these methods are (not) used as frequently as they should be.
Indeed, the existence of alternative methods for the assessment of pluripotency such as PluriTest and ScoreCard questions the great use of the teratoma assay for this goal. Even though we agree the comparison between this methods and the teratoma assay is interesting (other original research papers have been published about the topic), a review on this matter was beyond the scope of our systematic analysis, as here we wanted to emphasize the little standardization the teratoma assay presents. Nonetheless, in order to highlight this issue, we have added the following text in the discussion/after line 429
“It is remarkable how, since the publications of Scorecard and PluriTest (2011 and 2015 respectively), only 50-60 articles have been published reporting their use for pluripotency assessment (numbers extracted from the same search engines used in our study). This represents an estimated mere 10-20% compared to the articles that were published since then reporting the use of the teratoma assay, further highlighting the little adherence to these alternative animal-free approaches.”
3) English should be corrected throughout the article – I would recommend a careful checking as some sentences are not clear enough or hard to follow (focus particularly on the Abstract section). Here are some examples/typos/suggestions to correct:
-lines 15-16: the sentence is missing a verb?
The sentence has been changed to ‘In reporting new human pluripotent stem cell lines, their clonal derivatives or the safety of differentiated derivatives for transplantation, assessment of pluripotency is essential.’ We have also performed a full text English proofreading of the article.
-line 44: “are” instead of “be”
Changed as suggested
-line 45: should be reformulated (by assays determined by the ultimate… ?)
Changed as suggested
-line 63: “different” seems to be over-abundant
As suggested ‘different’ has been removed
-line 68: “stem cells” or “cell lines”
As suggested ‘stem cell is’ has been changed to ‘stem cells are’.
-line 73: genes or proteins
As suggested ‘gene or protein’ has been changed to ‘genes and proteins’.
-line 80: “is” instead of “be”
Changed as suggested
-line 87: 10 essential requirements
Changed as suggested
-line 196: “just” should be deleted
We have kept the word ‘just’ since we are of the opinion that it is important to emphasize the short time period
-line 198: duplicates
Duplicate is correct
-line 270: common (or better “frequent”?)
Commonly has been changed to frequent
-line 274: … for injection. (“site” should be deleted)
Changed as suggested
-line 277: “was” instead of “were”
Changed as suggested
-line 317-318: … a number of alternative … reduce the number of …
Changed as suggested
-line 335: “Changes in teratoma experiments over the years” is not clear why present here
This sentence has been removed, as suggested
-line 349: delete “there”
Changed as suggested
-line 364: … for at least seven …
Changed as suggested
-line 436: among
Amongst is correct, we have not changed this
Round 2
Reviewer 1 Report
There are some disagreements in the authors explanations, i.e. they cite a direction “The same directive acknowledges that animals have an intrinsic value and clearly states that no animal experiment should be performed for scientific or educational purposes if these can be reached with animal-free methods (EU Directive 2010/63/EU/art. 12).” However, at the end the authors say “…to aim to develop and embrace in vitro alternatives”, which means that so far there are no teratoma alternatives in vitro or at least the effective ones that could be trusted.
The chapter of in vitro teratoma studies (at what stage they are, what type, some examples, their benefits and drawback compare to the in vivo studies and other) is needed to better understand today's achievements, possibilities and/or choices.
The information about the main directions of iPS differentiation directions in which iPSs are mostly studies and have most promising results also would give more scientific value and confirmation why do we need the iPS at all.
Now it is just a critic written to the animal studies and statistical calculations or articles without strong enough scientific value, which makes this review more sociological than scientific.
The authors should not forget that cancer and drug investigation studies use considerably higher numbers of experimental animals in their studies than the iPS teratoma studies.
Author Response
Reviewer 1
There are some disagreements in the authors explanations, i.e. they cite a direction “The same directive acknowledges that animals have an intrinsic value and clearly states that no animal experiment should be performed for scientific or educational purposes if these can be reached with animal-free methods (EU Directive 2010/63/EU/art. 12).” However, at the end the authors say “…to aim to develop and embrace in vitro alternatives”, which means that so far there are no teratoma alternatives in vitro or at least the effective ones that could be trusted.
Thank you for the comment; we have better clarified in lines 429-448 the reason why PluriTest and ScoreCard can be used as in vitro assays for testing pluripotency while still the teratoma assay is the only in assay to give information regarding malignancy. We hope that in this latest version that recommendations of the Internation Stem Cell initiative-ISCI (ref. 3) have been presented in a more clear and exhaustive manner.
The chapter of in vitro teratoma studies (at what stage they are, what type, some examples, their benefits and drawback compare to the in vivo studies and other) is needed to better understand today's achievements, possibilities and/or choices.
From line 436 to 446 we have tried to explain in a more clear way that the teratoma assay gives information of malignancy at a histological level and that anyway certain features of malignancy also in the teratoma assay need to be further investigated.
The information about the main directions of iPS differentiation directions in which iPSs are mostly studies and have most promising results also would give more scientific value and confirmation why do we need the iPS at all.
We appreciate this comment and this consideration, but we think that this is outside the scope of our systematic review.
Now it is just a critic written to the animal studies and statistical calculations or articles without strong enough scientific value, which makes this review more sociological than scientific. The authors should not forget that cancer and drug investigation studies use considerably higher numbers of experimental animals in their studies than the iPS teratoma studies.
We did not mean to position our studies as a blunt critic to animal studies; we believe as many others at the scientific and governmental levels, that animal models have been incredibly important for further advance of science but we think that it is ethically responsible to promote animal-free assays especially when the in vitro assays have been validated and standardized. In this case we would like to further promote the validates assay for pluripotency since the global study by the ISCI and other studies support the reliability of these assays. In this latest proposed version of the manuscript (line 429-448), we further highlight that to date there is not a vitro assay available for testing malignancy. We acknowledge this point, and we suggest further research related to the very complex malignancy aspect.

Reviewer 2 Report
For a systematic review, there is evidence selection bias. If the authors do not identify all available data on the topic, this bias occurs. Although the sample of articles included is representative of the published papers where the teratoma assay is used, this subset does not reflect the entire evidence base. If there is a possibility of other articles not mentioning “Teratoma assay” in the title or abstract that did perform the assay and thus were left out of the screening procedure, I think the authors should mention this potential limitation in the discussion at least.
Author Response
Reviewer 2
For a systematic review, there is evidence selection bias. If the authors do not identify all available data on the topic, this bias occurs. Although the sample of articles included is representative of the published papers where the teratoma assay is used, this subset does not reflect the entire evidence base. If there is a possibility of other articles not mentioning “Teratoma assay” in the title or abstract that did perform the assay and thus were left out of the screening procedure, I think the authors should mention this potential limitation in the discussion at least.
Thank you for further stressing this point; we agree that it is fair to include this point in the discussion (see lines 376-383) as a possible limitation.
